# Cytotoxic, Anti-Hemolytic, and Antioxidant Activities of *Ruta chalepensis* L. (Rutaceae) Extract, Fractions, and Isolated Compounds

**DOI:** 10.3390/plants12112203

**Published:** 2023-06-02

**Authors:** Joel H. Elizondo-Luévano, Nancy E. Rodríguez-Garza, Aldo F. Bazaldúa-Rodríguez, César I. Romo-Sáenz, Patricia Tamez-Guerra, María J. Verde-Star, Ricardo Gomez-Flores, Ramiro Quintanilla-Licea

**Affiliations:** 1Departamento de Química, Facultad de Ciencias Biológicas, Universidad Autónoma de Nuevo León (UANL), San Nicolás de los Garza 66455, NL, Mexico; joel.elizondolv@uanl.edu.mx (J.H.E.-L.); aldo.bazalduarg@uanl.edu.mx (A.F.B.-R.); maria.verdest@uanl.edu.mx (M.J.V.-S.); 2Grupo de Enfermedades Infecciosas y Tropicales (e-INTRO), Instituto de Investigación Biomédica de Salamanca, Centro de Investigación de Enfermedades Tropicales de la Universidad de Salamanca (IBSAL-CIETUS), Facultad de Farmacia, Universidad de Salamanca (USAL), 37007 Salamanca, Spain; nancy.rodriguezg@usal.es; 3Departamento de Microbiología e Inmunología, Facultad de Ciencias Biológicas, (UANL), San Nicolás de los Garza 66455, NL, Mexico; cesar.romosnz@uanl.edu.mx (C.I.R.-S.); patricia.tamezgr@uanl.edu.mx (P.T.-G.)

**Keywords:** chalepensin, cytotoxic activity, ethnopharmacology, furanocoumarins, graveoline, hemolysis, rutamarin

## Abstract

*Ruta chalepensis* is an herb used to treat various ailments, and its potential cytotoxic effects on different tumor cell lines have been extensively studied. The present study aimed to evaluate the cytotoxic activity of *R. chalepensis* methanol extract (RCME), sub-partitions obtained from solvents of increasing polarity, and major compounds, as well as their hemolytic, anti-hemolytic, and antioxidant potential. The in vitro cytotoxic activity against the human hepatocarcinoma (HEP-G2) and the murine lymphoma cell line (L5178Y-R) was evaluated using the colorimetric 3-(4,5-dimethylthiazol-2-yl)-2,5-diphenyltetrazolium bromide (MTT) reduction assay, whereas selectivity indices (SIs) were determined by comparing cytotoxicity against normal African green monkey kidney cells (VERO) and human peripheral blood mononuclear cells (PBMC). Hemolytic and anti-hemolytic activities were evaluated on human erythrocytes. The most effective cytotoxic treatment was evaluated for nitric oxide release by J774A.1 macrophages. Antioxidant activity of *R. chalepensis* material was also determined. Results showed that RCME produced significant (*p* < 0.05) cytotoxicity in HEP-G2 (IC_50_ = 1.79 µg/mL) and L5178Y-R (IC_50_ = 1.60 µg/mL) cells and exhibited high SIs (291.50 and 114.80, respectively). In addition, the *n*-hexane fraction (RCHF) showed an IC_50_ of 18.31 µg/mL in HEP-G2 cells and an SI of 9.48 in VERO cells, whereas the chloroform fraction (RCCF) evidenced an IC_50_ of 1.60 µg/mL in L5178Y-R cells and an SI of 34.27 in PBMC cells. Chalepensin (CHL), rutamarin (RTM), and graveolin (GRV), which are major components of *R. chalepensis*, showed high activity against L5178Y-R cells, with IC_50_ of 9.15, 15.13 and SI of 45.08 µg/mL, respectively. In addition, CHL, RTM, and GRV showed SIs of 24.76, 9.98, and 3.52, respectively, when compared with PBMC cells. RCME at concentrations of 125 µg/mL and 250 µg/mL, significantly (*p* < 0.05) decreased nitrite production in J774A.1 cells, when exposed to lipopolysaccharide. This study demonstrated that RCME showed significant cytotoxic activity against HEP-G2 and L5178Y-R cells, without affecting normal VERO, PBMC, and J774A.1 cells.

## 1. Introduction

Cancer is one of the leading causes of human mortality and morbidity in the 21st century, with one in four people at risk of developing cancer in their lifetime [1]. Current alternatives for cancer therapy that are available and effective are limited and often produce severe toxic and side effects [2]. Thus, searching for new, effective, safe, and affordable anticancer drugs is part of many significant modern drug discovery efforts worldwide. Plants have been studied for their potential antitumor effects [3]. *Ruta chalepensis* L. (Rutaceae), also known as rue or ruta, is a medicinal plant native to the Mediterranean region used for its medicinal properties globally. In Mexico, this plant is used by indigenous communities to treat gastrointestinal diseases [4], and studies have proven its cytotoxic, anti-hemolytic, antiparasitic, anti-inflammatory, and abortive activities [5,6,7,8,9]. *R. chalepensis* possesses different natural metabolites that are bioactive, which are distributed in leaves, stems, and roots, such as alkaloids (graveoline), coumarins (rutamarin), flavonoids, and furanocoumarins (chalepensin) [6,10,11,12].

There are several studies on *Ruta* species that describe the pharmaceutical properties associated with their bioactive compounds [12]. The ethanol extract of *R. chalepensis* and the flavonoid rutin present in *R. chalepensis* protect red blood cells from oxidative stress induced by radicals, such as 2,2′-azobis (2-methylpropionamidine) dihydrochloride (AAPH), in patients with colon cancer [13]. In an in vivo mouse model, *R. chalepensis* ethanol extract was shown to possess potent immunopharmacological properties that counteracted the lethal effects of high doses of lipopolysaccharide (LPS) [14]. Furthermore, the antiplatelet activity of ethyl acetate (EtOAc) and methanol extracts of *R. chalepensis*, as well as chalepensin (CHL) and bergaptene from this plant were studied in vitro in human blood [15]. 

There are a variety of studies where the extracts, fractions, and major components of plants of the *Rutaceae* family, showed antioxidant and cytotoxicity against a variety of tumor cells [12,16].To date, there are several studies with furanocoumarins, including CHL (also known as xylotenin) and rutamarin (RTM; also known as chalepin acetate), that report cytotoxic, antiproliferative, antimicrobial, antiparasitic, anti-inflammatory, and anticancer properties, among others [6,15,17,18,19,20]. Another important family of compounds are the alkaloids, such as graveoline (GRV), which possess antibacterial, antifungal, cytotoxic, apoptotic, and autophagic activity [21,22,23]. However, to date, there are no reports of systematic preclinical or clinical trials of these compounds in humans.

We have previously identified *R. chalepensis* main components, including CHL, RTM, and GRV [6,24,25] (Figure 1) and reported the in vitro cytotoxicity of *R. chalepensis* methanol extract (RCME) on human hepatocellular carcinoma (HEP-G2) cells [5] and murine L5178Y-R lymphoma cells [26]. Therefore, the aim of the present study was to evaluate the cytotoxic and antihemolytic activities of *R. chalepensis* methanol extracts, sub-partitions obtained with solvents of increasing polarity (*n*-hexane, chloroform, and methanol), and major isolated compounds (CHL, RTM, and GRV). In addition, we investigated nitric oxide (NO) release and antioxidant potential induced by the treatment with the highest HEP-G2 and L5178Y-R cytotoxic activity.

## 2. Results and Discussion

### 2.1. Phytochemical Data and Biodirected Isolation of Major Ruta chalepensis Compounds

We evaluated *R. chalepensis* crude methanol extract (RCME), partitions obtained with solvents of increasing polarity, and the main components of *R. chalepensis* to broadly comprehend the biological activity of this plant. Advantages of partitioning plant extracts with solvents of different polarities and evaluating their in vitro biological studies have been demonstrated [27,28]. Table 1 shows the percentage of extraction yield of *R. chalepensis* extract, sub-partitions, and pure compounds. Yield percentages for the secondary metabolites and the chloroform (CHCl_3_) fraction were <1% (*w*/*w*).

We have previously reported the biodirected isolation of *R. chalepensis* secondary metabolites [6,24,25], which were identified using spectroscopy and spectrometry, and compared with bibliographic data (spectroscopic data of CHL, RTM, and GRV are available as Appendix A). Details of the methodological steps for this biodirected isolation of *R. chalepensis* compounds are available in the Appendix A. To purify the main metabolites of *R. chalepensis*, RCME was separated into different components using *n*-hexane extraction, and the residue produced exhibited significant activity against HEP-G2 and L5178Y-R tumor cells. Subsequent chromatography of this residue on a silica gel column resulted in the isolation of CHL. Additional processing of the methanol residue involved partitioning between MeOH and ethyl EtOAc, followed by chromatography of the EtOAc residue on a silica gel column, and yielded RTM and GRV of high purity.

Low yields of secondary metabolites may be due to the extraction method and solvents used in our research, as previous studies using subcritical fluid extraction with CO_2_ showed significant number of components, even more than those reported in rue extracts using dichloromethane and steam distillation [29]. Studies found 6.6% CHL, 7.1% chalepin, 5.5% bergapten, and 0.85% GRV in leaf extracts, whereas in flower extracts, 10.8% CHL, 9.1% RTM, 9.0% psoralen, 7.1% bergapten, and 0.68% GRV were found, and 13% CHL was found in stem extracts. Moreover, there may also be a difference in the concentration of secondary metabolites, depending on the part of the plant used, as well as in varieties or subvarieties belonging to the same family in different geographical areas. In this regard, in the essential oil of *Ruta graveolens* from the West Romanian area, metabolites, such as 2-undecanone (76.19%), 2-nonanone (7.83%), 2-undecanol (1.85%), 2-decanone (0.75%), and bergapten (0.56%), have been found [30]. In addition, other studies reported ketones (2-undecanone, 2-decanone, and 2-dodecanone) that represent >80% of *R. chalepensis* components [7].

### 2.2. Cytotoxic Activity

Targeted triggering of tumor cell death is the therapeutic goal of recent cancer treatments. For this purpose, many medicinal plants and plant-derived products are being tested [31,32]. Results of the effect of *R. chalepensis* extracts, fractions, and isolated compounds on tumor and normal cells toxicity, as well as selectivity indices (SIs) are shown in Table 2. The cell response of adherent HEP-G2 cells was compared with that of Vero cells, whereas the response of non-adherent L5178Y-R cells was compared with that of human PBMC. RCME was most effective against tumor cell lines, showing the highest SIs (291.50 on HEP-G2 cells and 114.80 on L5178Y-R cells) compared with other treatments (*p* < 0.001), whereas the least effective treatment was RCMF, which evidenced the lowest SIs. *R. chalepensis* is well known for its uses in traditional medicine. Some of its main components, such as furanocoumarins (chalepine, CHL, and RTM) and alkaloids (GRV), have been evaluated in different countries and their bioactivities have been demonstrated. However, we lack reports on the preclinical testing of these compounds [33,34,35].

*n*-Hexane partition (RCHF) from RCME resulted in a residue with high activity against HEP-G2 tumor cells (IC_50_ = 18.31 µg/mL) compared to the other fractions (*p* < 0.01). RCCF showed high activity against L5178Y-R lymphoma cells (IC_50_ = 86.14 µg/mL) compared to other treatments (*p* < 0.001). However, the CHCl_3_ partition presented an IC_50_ of 36.60 µg/mL against HEP-G2 cells and an IC_50_ of 1.60 µg/mL against L5178Y-R cells (*p* < 0.001). Furanocoumarins, CHL and RTM, and the quinoline alkaloid, GRV, were effective against L5178Y-R cells as they showed a significant growth inhibition effect against these cells (*p* < 0.01) with IC_50_ less than 50 µg/mL and Sis higher than 3.5, indicating their high antitumor potential [5].

Sis of CHL, RTM, and GRV were higher than 3 (lower limit proposed in this research), indicating low toxicity of CHL, RTM, and GRV against normal cells but significant tumor cell growth inhibition [36]; these compounds showed significant higher Sis against L5178Y-R cells compared to HEP-G2 cells (*p* < 0.05). Anti-inflammatory, anticancer, antiviral, antiparasitic, and amoebicidal activities of these compounds have been documented [6,33,37]. However, most of the studies on the bioactivity of these compounds have been predominantly conducted in vitro and only a few conducted in vivo and in silico [22,38]. A study evaluating extracts with solvents of different polarity from *Ruta angustifolia* showed cytotoxic effect against different tumor lines, highlighting the cytotoxic effect of MeOH and CHCl_3_ extracts in addition to the cytotoxic effect of different metabolites, such as furanocoumarins (chalepin, CHL, and RTM) and alkaloid (GRV), against human lung carcinoma cells (A549) [39]. A previous study demonstrated that GRV, the principal constituent of *R. graveolens*, induced both apoptotic and autophagic ROS-mediated cell death in cutaneous melanoma cells, a desirable quality for designing effective anticancer drugs [21], and may support the activity shown by CHL, RTM, and GRV against both tumor cell lines evaluated in this research.

### 2.3. Evaluation of Hemolytic and Anti-Hemolytic Activities

Pharmaceutical industries utilize in vitro hemolytic and anti-hemolytic (AAPH) assays to develop new pharmaceuticals and confirm the cytotoxicity of an extract or new drug in mammalian cells, and they are essential to generate reliable biological evidence [40]. Thus, we evaluated erythrocyte hemolytic and anti-hemolytic effects using the most effective extract against HEP-G2 and L5178Y-R tumor cells. Table 3 shows the hemolytic activity of the different treatments that were previously evaluated against tumor and normal cells. Based on the criteria of Chomchan et al. (2018), in the drug screening system, half-maximal inhibitory concentrations (IC_50_) higher than 90 µg/mL were classified as non-toxic [41]. Thus, all treatments, except GRV, showed low hemolytic activity with IC_50_ higher than 100 µg/mL (*p* < 0.001 compared to other treatments), demonstrating low toxicity to red cells (Table 3).

The AAPH radical-induced hemolysis protection assay is utilized as an ex vivo model to demonstrate the exceptional antioxidant potential of certain natural products as it causes lipid peroxidation in normal red blood cells [42]. Assessment of anti-hemolytic activity revealed that RCMF and RCME had the highest activity, with IC_50_ values of 9.33 µg/mL and 28.29 µg/mL, respectively, whereas RTM and GRV did not exhibit any anti-hemolytic activity (*p* < 0.001) (Table 3). Protective activity may be attributed to the polyphenol content as polyphenols interact with the components of the erythrocyte membrane through hydrogen bonds and prevent the oxidation of membrane proteins and lipids [43]. This is directly related to the antioxidant potential due to the possible composition of the methanolic extract and its partitions, highlighting some phenolic compounds, flavonoids, and furanocoumarins [5,44,45,46].

### 2.4. Nitrite Production Assay

We evaluated the in vitro effect of RCME and RCME + 2 µg/mL LPS on nitrite release (as a measure of nitric oxide production) using J774A.1 murine macrophages. Table 4 shows murine macrophage viability along with nitrite levels. A significant (*p* < 0.05) decrease in J774A.1 cells viability was observed after RCME treatment in a concentration-dependent manner, probably because of nitric oxide in these cells, which has been reported to induce murine macrophage apoptosis (20% cytotoxicity) at about 10 μM nitrites [34]. The capacity of inhibition of NO production caused by RCEM is shown in Table 4. When the inflammatory inducer LPS was used, no significant differences were found at concentrations of 15.62, 31.25, 62.50, and 500 µg/mL (*p* > 0.05), but significant differences were observed at concentrations of 125 and 250 µg/mL in comparison with the control (*p* < 0.001). IC_50_ values for RCME and RCME + LPS treatments were 400.78 and 238.13 µg/mL, respectively (*p* < 0.01).

Cytotoxicity assays were based on Lopez Villarreal et al. [47], considering the following parameters: (a) non-cytotoxic = 75% to 100% viability, (b) slightly cytotoxic = 50% to 74% viability, (c) moderately cytotoxic = 25% to 49% viability, and d) highly cytotoxic = 0% to 24% viability. Furthermore, a previous study with mice showed that *R. chalepensis* extract possessed potent immunopharmacological properties that counteracted the lethal effects of high doses of LPS in vivo and decreased NO production [14]. Therefore, our results showed that RCME and RCME + LPS effects ranged from non-cytotoxic to moderately cytotoxic.

Inducible nitric oxide synthase (iNOS) produces nitric oxide (NO) from L-arginine during the inflammatory process. The reaction between superoxide and NO results in the formation of peroxynitrite, a toxic substance causing tissue injury in inflammatory diseases [48]. It has been reported that phenolic compounds in natural extracts play a role in LPS-induced inhibition of NO in murine macrophages [41]. Overproduction of NO causes tissue damage and is associated with chronic inflammation [49]. Therefore, it was hypothesized that RCEM possesses immunomodulatory activity in response to LPS-induced inflammation. However, further confirmatory studies are required.

### 2.5. Antioxidant Activity

We evaluated the most effective cytotoxic treatment (RCME) to determine its antioxidant activity. Results showed it had potential to capture 1,1–diphenyl–2–picryl hydrazyl (DPPH) and 2,2′–azinobis–3–ethylbenzothiazoline–6–sulfonic acid (ABTS) radicals, and evidenced antioxidant activity with IC_50_ values of 89.90 µg/mL and 130.06 µg/mL, respectively (Table 5). Previous studies using *R. chalepensis* extracts have demonstrated antioxidant and chelating effects [50], which agrees with our present results. The antioxidant potential and the significant in vitro antiproliferative activity against human colorectal HT-29 and ovarian OV2008 cancer cell lines are attributed to the high content of diverse phenolic compounds and flavonoids present in crude extracts [51].

Szewczyk et al. [52] reported that *R. chalepensis* and *R. graveolens* MeOH extracts exhibited antioxidant activity with IC_50_ values of 1.67 mg/mL and 1.88 mg/mL, respectively, which was also evaluated by the DPPH uptake method, whereas Fakhfakh et al. [53] demonstrated IC_50_ values of 0.12 mg/mL and 0.22 mg/mL for *R. chalepensis* ethanol and aqueous extracts, respectively. These results are not in agreement with ours, as we found higher antioxidant activities. However, our observations agreed with those obtained by Rached et al. and others [37,54], who reported a IC_50_ values of 61.41 μg/mL for ethanol extract and 60.20 μg/mL for *R. chalepensis* methanol extract, which demonstrated the DPPH radical scavenging potential of *R. chalepensis*. Plants possess different bioactive effects, including antioxidant, anti-inflammatory, antibacterial, and fungicidal activities. They may also reduce toxins, such as hydrogen peroxide, malondialdehyde, and NO, which protect various organs, such as the kidney and liver, among others [55].

## 3. Materials and Methods

### 3.1. Ethical Statement

The procedures used in this study were approved by the Ethics Committee of the Universidad Autónoma de Nuevo León (UANL), registration no. CI-04-26-2022 (Appendix A). The experiment involving human erythrocytes was conducted with the informed consent of a healthy donor in compliance with the Official Mexican Technical Standard NOM-253-SSA1-2012 (Appendix A).

### 3.2. Chemicals and Reagents

All chemicals and solvents were of analytical grade. We obtained 2,2′-azino-bis(3-ethylbenzothiazoline-6-sulfonic acid) (ABTS), 2,2′-azobis (2-methylpropionamidine) dihydrochloride (AAPH), 2,2-diphenyl-1-picrylhydrazyl (DPPH), 3-(4,5-dimethylthiazol-2-yl)-2,5-diphenyltetrazolium bromide (MTT), ascorbic acid (vitamin C), chloroform (CHCl_3_), deuterochloroform (CDC_l3_), dimethyl sulfoxide (DMSO), ethyl acetate (EtOAc), potassium persulfate (K_2_S_2_O_8_), ferric chloride, Griess reagent, lipopolysaccharide (LPS) from *Escherichia coli* O26:B6 (smooth strain), *n*-hexane (Hexane), methyl alcohol (MeOH), RPMI-1640 medium, Sephadex^®^ LH-20, sodium bicarbonate (NaHCO_3_), sodium chloride (NaCl), sodium hydroxide (NaOH), sodium phosphate dibasic (Na_2_HPO_4_), sodium phosphate monobasic (NaH_2_-PO_4_), and sulfuric acid from Sigma-Aldrich^®^ (St. Louis, MO, USA) and 4-2(2-hydroxyethyl)-1-piperazine ethane sulfonic acid (HEPES) from Invitrogen™ (Waltham, MA, USA). Dulbecco’s modified Eagle medium (DMEM), 1% antibiotic/antimycotic solution, fetal bovine serum (FBS), and sodium bicarbonate (NaHCO_3_) were purchased from Gibco™ (Grand Island, NY, USA). Vincristine sulfate (VS) salt was acquired from Hospira Inc. (Lake Forest, IL, USA).

### 3.3. Plant Material and Extraction

Plants used in this study were purchased from Pacalli, Herbolaria Científica (https://www.pacalli.com/; accessed on 11 November 2022), a local supplier in Monterrey City, Nuevo Leon, Mexico. The plant was taxonomically identified as *Ruta chalepensis* L. by a curator from the herbarium of the Facultad de Ciencias Biológicas (FCB) of the UANL and assigned the voucher number FCB-UNL 30,654 (the identification document is attached as Appendix A). The botanical name and family of *R. chalepensis* have been taxonomically validated using the ThePlantList website (http://www.theplantlist.org; accessed on 1 November 2022).

*R. chalepensis* crude methanol extract (RCME) was prepared as previously reported [5]. To prepare the plant extract, 100 g of the aerial part of the plant material (dried and milled) was treated with 1000 mL of MeOH in a Soxhlet apparatus for 48 h [6]. Resulting soluble partitions were then obtained using the solvents of increasing polarity hexane (RCHF), chloroform (RCCF), and MeOH (RCMF), in a Soxhlet apparatus for 48 h. Extracts and fractions were filtered, concentrated using vacuum evaporation with a rotary evaporator (Buchi R-3000; Brinkman Instruments, Inc., Westbury, NY, USA) at 80 rpm and 40 °C in a water bath, and stored at 4 °C in amber bottles, until use [56]. Extraction yield percentages were calculated using the following Formula (1):(1)$$\%\;Yield=\frac{Final\;weight}{Initial\;weight}\times 100$$

In addition, we showed the cytotoxic activity of the main components chalepensin (CHL), rutamarin (RTM), and graveoline (GRV), which we previously from *R. chalepensis* [6,24,25]. We reported the isolation of CHL in DOI: 10.3390/molecules191221044 [24], RTM in DOI: 10.3390/molecules26123684 [6], and GRV in DOI: 10.1055/s-0036-1596528 [25], and the identification was performed by spectroscopy and spectrometry with a Bruker Spectrometer (Model Advance DPX400, 9.4 Teslas; Bruker Corporation, Billerica, MA, USA) and compared with bibliographic data. Spectroscopic analysis data of CHL, RTM, and GRV are available as Appendix A.

### 3.4. Cell Viability Assay

The human hepatocellular carcinoma cell line HEP-G2 (ATCC HB-8065) and the monkey kidney epithelial cell line VERO (ATCC CCL-81) were obtained from the Laboratory of Immunology and Virology in FCB at UANL (FCB-UANL), México, and cultured in DMEM, supplemented with 10% heat-inactivated FBS, 2% NaHCO_3_, and HEPES [57]. Murine lymphoma L5178Y-R cells (ATCC CRL-1722) and human peripheral blood mononuclear cells (PBMC) were maintained in RPMI-1640 culture medium, supplemented with 10% heat-inactivated FBS and 1% antibiotic/antimycotic solution [26]. Assays were conducted with adherent HEP-G2 and VERO cells on flat-bottom microplates and non-adherent L5178Y-R and PBMC cells were evaluated on round-bottom microplates.

Cells were cultured at 37 °C and 5% CO_2_ in 95% air in a humidified incubator, in the presence or absence of treatments for 72 h. After which they were incubated with MTT (0.5 mg/mL final concentration) at 37 °C in 5% CO_2_ for 3 h and supernatants were discarded after the incubation period. Resulting formazan crystals were dissolved in DMSO, and optical densities (OD) were measured at 540 nm, using a microplate reader (Molecular Devices Corporation in Palo Alto, CA). In addition, maximal inhibitory concentration (IC_50_) values were determined after 72 h incubation with treatments at concentrations ranging from 7.81 µg/mL to 1000 µg/mL and ODs were measured at 570 nm on a microplate reader (Molecular Devices Corporation) [5]. The positive control consisted of 0.05 µg/mL VS and the negative control was culture medium alone [58]. All treatments were diluted in DMSO to a final well test concentration not exceeding 0.5% (*v*/*v*). Cell viability was determined using the following Formula (2):(2)$$Cell\;viability\;\%=\frac{{OD}_{570nm}Treatment}{{OD}_{570nm}Negative\;control}\times 100$$

### 3.5. Selectivity Index

To determine the selectivity index (SI) for each treatment, we divided the IC_50_ of tumor cells by that of normal cells. SI was assessed for adherent cells (non-tumor/tumor) and for non-adherent cells (non-tumor/tumor). A value higher than 3 was considered to possess high selectivity [59]. SI was calculated using the following Formula (3):(3)$$SI=\frac{{IC}_{50}\;Normal\;Cells}{{IC}_{50}\;Tumor\;Cells}$$

### 3.6. Hemolytic Activity Test

The hemolytic activity was assessed using the hemolysis test [60] at treatment concentrations ranged from 7.81 µg/mL to 1000 µg/mL in PBS at pH 7.4. IC_50_ values were defined as the sample concentration needed to cause 50% hemolysis of human red blood cells. The percentage of hemolysis was determined by measuring ODs at 540 nm as follows (4):(4)$$Hemolysis\;\%=\frac{{OD}_{540nm}\;Treatment}{{OD}_{540nm}\;Positive\;control}\times 100$$

### 3.7. Anti-Hemolytic Activity Test

The AAPH inhibition test was used to determine the anti-hemolytic activity, as previously reported [61]. Hemolysis was induced by the AAPH radical (150 mM in PBS) as a positive control. Treatment concentrations ranged from 7.81 µg/mL to 1000 µg/mL in PBS at pH 7.4 plus AAPH. IC_50_ values were defined as the sample concentration needed to cause 50% hemolysis of human red blood cells. The anti-hemolytic activity was evaluated using the following Formula (5):(5)$$\%\;Anti-hemolytic\;Activity=1-\left(\frac{{OD}_{570nm}\;Treatment}{{OD}_{570nm}\;Positive\;control}\right)\times 100$$

### 3.8. Nitrite Determination

Accumulation of nitrite in the supernatants of J774A.1 macrophage cultures indicated nitric oxide production by resident or activated cells [56]. Untreated and LPS-treated macrophages were incubated in triplicate at 37 °C in 5% CO_2_ for 24 h at concentrations ranging from 15.62 µg/mL to 500 µg/mL of the most effective treatment (RCME) against tumor cells in RPMI-1640 culture medium, supplemented with 10% FBS and 1% antibiotic/antimycotic solution. After incubation, supernatants were obtained, and nitrite levels were determined with the Griess reagent, as reported elsewhere, using NaNO_2_ as a standard [62]. ODs at 540 nm were then determined in a microplate reader (Molecular Devices Corporation). Cell viability was also evaluated using the MTT reduction assay, as explained above.

### 3.9. Antioxidant Activity

We determined the antioxidant activity of RCME, using the DPPH radical and ABTS radical scavenging methods. Antioxidant activity (free radical scavenging potential) was expressed as IC_50_ in µg/mL [37] and was defined as the concentration of the test material required to produce a 50% decrease in the initial concentration of DPPH or ABTS, using vitamin C as a positive control [63].

#### 3.9.1. DPPH Radical-Scavenging Assay

We mixed 100 μL of treatments (3.9 µg/mL to 500 µg/mL) and 100 μL of 0.1 mM DPPH methanol solution in 96-well flat-bottom microplates (Corning Inc., Corning, NY, USA). Microplates were incubated in the dark at room temperature for 30 min, and the OD was recorded at 517 nm. Percentage inhibition of DPPH was calculated using the following Formula (6):(6)$$DPPH\;scavenging\;\%=\frac{{OD}_{517}\;Negative\;control-{OD}_{517}\;Treatment}{{OD}_{517}\;Negative\;control}\times 100$$

#### 3.9.2. ABTS Radical-Scavenging Assay

We used the method described by Khlifi et al. [64] to determine the potential of RCME to scavenge the radical ABTS. The assay was performed in a 96-well flat-bottom microplate. ABTS was prepared by combining 7 mM ABTS (5 mM NaH_2_-PO_4_, 5 mM Na_2_HPO_4_, and 154 mM NaCl; pH 7.4) with 2.5 mM K_2_S_2_O_8_ (final concentration) in distilled water and kept in darkness at room temperature for 14 to 16 h, until use. ABTS was then diluted with MeOH to obtain an absorbance of 0.70 ± 0.05 at 734 nm. RCME concentrations ranging from 3.9 µg/mL to 500 µg/mL were evaluated, and 10 µL of the extract was mixed with 90 µL of ABTS solution in a 96-well flat-bottom microplate (Corning) and incubated at room temperature in darkness for 6 min, after which ODs were measured at 734 nm [64]. The percentage inhibition of ABTS was calculated using the following Formula (7):(7)$$ABTS\;scavenging\;\%=\frac{{OD}_{734nm}\;Negative\;control-{DO}_{734}\;Treatment}{{OD}_{734nm}\;Negative\;vontrol}\times 100$$

### 3.10. Statistical Analysis

We used the Graph Pad Prism 6 software (GraphPad Software Inc., San Diego, CA, USA) for statistical analyses. Results were shown as mean ± SD of triplicate determinations from three independent experiments. A one-way analysis of variance was employed to determine the significant difference between concentrations. In addition, the *post-hoc* Tukey test was utilized to evaluate the difference between treatment means. IC_50_ values were calculated using the Probit test.

## 4. Conclusions

We demonstrated that *Ruta chalepensis* crude methanol extract, fractions, and its major components chalepensin (CHL), rutamarin (RTM), and graveolin (GRV), possessed significant in vitro cytotoxic activities against HEP-G2 and L5178Y-R cells, with high selectivity indices (SIs), as compared with normal VERO cells and PBMC. The *R. chalepensis* methanol extract RCME was particularly effective against tumor cells, without affecting normal cells, with a SI of 291.50 for adherent cells and 114.80 for non-adherent cells. This may be attributed to the synergistic effect of various phytochemical compounds of the plant. In addition, RCME decreased nitrite production in resident and LPS-activated J774A.1 macrophages, and reduced their viability in a concentration-dependent manner (probably due to NO [34]).

Compounds CHL, RTM, and GRV showed significant cytotoxic effects against L5178Y-R cells. However, CHL exhibited high SIs against HEP-G2 (SI = 5.08) and L5178Y-R (SI = 24.76) tumor cells, without affecting normal cells. RTM and GRV showed no statistical difference (*p* > 0.05) of SIs, when evaluated against HEP-G2 cells. CHL and RTM did not show significant hemolytic activity, supporting their potential use in future studies.

Based on current findings, CHL, RTM, and GRV compounds may be utilized as antiproliferative biomolecules against tumor cells and to develop potential antitumor therapeutic agents. Therefore, it is necessary to scale-up cytotoxic compounds, using chemical synthesis to study them in-depth and demonstrate their therapeutic effects. We will further investigate the antitumor potential of *R. chalepensis* extracts, fractions, and major components to treat cancer, such as HEP-G2 hepatocellular carcinoma and L5178Y-R lymphoma, in preclinical models and elucidate the associated molecular mechanisms. 

## Figures and Tables

**Figure 1 plants-12-02203-f001:**
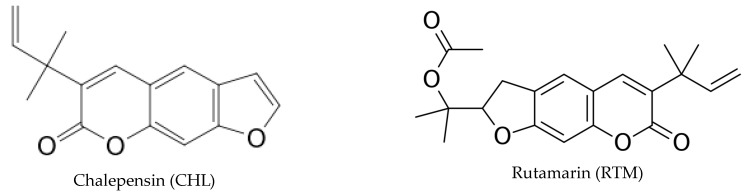
Structure of CHL (PubChem CID: 128834; https://pubchem.ncbi.nlm.nih.gov/compound/Chalepensin), RTM (PubChem CID: 26948; https://pubchem.ncbi.nlm.nih.gov/compound/26948), and GRV (PubChem CID: 353825; https://pubchem.ncbi.nlm.nih.gov/compound/353825). All structure links were accessed on 14 October 2022.

**Table 1 plants-12-02203-t001:** Extraction yield percentages.

Treatments	Abbreviation	% Extraction Yield
*R. chalepensis* MeOH extract	RCME	19.39
*R. chalepensis n*-hexane fraction	RCHF	1.55
*R. chalepensis* CHCl_3_ fraction	RCCF	0.90
*R. chalepensis* MeOH fraction	RCMF	7.73
Chalepensin	CHL	0.41
Rutamarin	RTM	0.42
Graveoline	GRV	0.03

**Table 2 plants-12-02203-t002:** Tumor and normal cells toxicity of *R. chelepensis* extracts, fractions, and isolated compounds.

Treatments	IC_50_ (µg/mL)		SI	IC_50_ (µg/mL)		SI
	VERO	HEP-G2		PBMC	L5178Y-R	
RCME	522.08 ± 29.96 ^e^	1.79 ± 0.38 ^a^	291.50 ^e^	183.91 ± 11.89 ^bc^	1.60 ± 0.02 ^a^	114.80 ^f^
RCHF	173.59 ± 9.83 ^b^	18.31 ± 1.58 ^b^	9.48 ^d^	225.23 ± 20.64 ^c^	86.14 ± 5.43 ^d^	2.61a ^b^
RCCF	84.37 ± 4.12 ^a^	36.60 ± 3.71 ^bc^	2.31 ^a^	54.83 ± 12.45 ^a^	1.60 ± 0.11 ^a^	34.27 ^e^
RCMF	1057.14 ± 12.16 ^f^	519.22 ± 20.32 ^f^	2.04 ^a^	338.57 ± 20.96 ^d^	311.52 ± 32.90 ^f^	1.09 ^a^
CHL	387.15 ± 9.24 ^c^	76.17 ± 2.85 ^d^	5.08 ^c^	226.46 ± 5.67 ^c^	9.15 ± 1.94 ^b^	24.76 ^d^
RTM	429.99 ± 4.07 ^d^	129.62 ± 3.68 ^e^	3.32 ^b^	151.03 ± 10.08 ^b^	15.13 ± 3.77 ^c^	9.98 ^c^
GRV	178.81 ± 7.39 ^b^	54.62 ± 3.34 ^c^	3.27 ^b^	158.84 ± 30.56 ^b^	45.08 ± 10.11 ^d^	3.52 ^b^
*p*—ANOVA	<0.01	<0.001	<0.01	<0.05	<0.001	<0.01

IC_50_ values (µg/mL) against tumor and normal cells are presented as means ± SD, with significant (*p* < 0.05) differences indicated by different letters within the same column, as determined using the post-hoc Tukey test. The selectivity index (SI) was calculated by dividing the IC_50_ of normal cells with the IC_50_ of tumor cell lines after 72 h of incubation and using 0.05 µg/mL vincristine sulfate as a positive control.

**Table 3 plants-12-02203-t003:** Hemolytic and anti-hemolytic activities.

Treatments	Hemolytic Activity IC_50_ (µg/mL)	Anti-Hemolytic Activity IC_50_ (µg/mL)
RCME	738.73 ± 20.74 ^c^	28.29 ± 2.31 ^b^
RCHF	824.90 ± 38.94 ^d^	930.94 ± 11.77 ^e^
RCCF	>1500 ^†^	843.56 ± 37.43 ^d^
RCMF	>1500 ^†^	9.33 ± 0.51 ^a^
CHL	101.07 ± 4.20 ^b^	179.86 ± 4.76 ^c^
RTM	>1500 ^†^	NA
GRV	57.60 ± 11.26 ^a^	NA
*p*—ANOVA	<0.001	<0.001

Data represent mean ± SD of IC_50_ values (µg/mL). Different letters within the same column are significantly (*p* < 0.05) different, analyzed by the post-hoc Tukey test. ^†^ IC_50_ values above 1500 µg/mL were not included in the Tukey test analysis. NA: No activity.

**Table 4 plants-12-02203-t004:** Murine macrophage viability, nitrite release, and IC_50_.

Treatments	RCME	RCME *+* LPS	RCME	RCME *+* LPS
(µg/mL)	% Viability		Nitrites (µM)	
0 (Control)	100.00 ± 0.09 ^e^	100.00 ± 0.27 ^d^	8.16 ± 1.37 ^a^	9.73 ± 1.49 ^ab^
15.62	92.68 ± 4.31 ^d^	89.73 ± 10.54 ^c^	8.18 ± 0.69 ^a^	10.91 ± 1.22 ^b^
31.25	90.34 ± 7.37 ^d^	86.82 ± 9.20 ^c^	8.39 ± 0.77 ^a^	10.59 ± 1.26 ^b^
62.5	88.30 ± 6.32 ^d^	84.56 ± 4.96 ^c^	8.54 ± 0.76 ^ab^	10.37 ± 1.83 ^b^
125	80.99 ± 10.46 ^c^	69.85 ± 9.09 ^b^	8.79 ± 1.37 ^ab^	9.23 ± 0.69 ^a^
250	58.46 ± 10.51 ^b^	43.59 ± 9.60 ^a^	9.00 ± 2.91 ^ab^	9.27 ± 0.48 ^a^
500	42.32 ± 9.90 ^a^	42.72 ± 4.88 ^a^	9.24 ± 1.74 ^b^	12.51 ± 1.67 ^c^
*p*—ANOVA	<0.01	<0.01	<0.05	<0.01
IC_50_ (µg/mL)	400.78 ± 51.05	238.13 ± 59.12		

Murine macrophages (J774A.1 cells) viability, IC_50,_ and nitrite production (µM) after RCME and RCME + LPS (2 µg/mL) treatments. Data represent means ± SD of triplicate determinations from three independent experiments. Different letters within the same column indicate a significant difference (*post-hoc* Tukey test).

**Table 5 plants-12-02203-t005:** RCME antioxidant activity.

Antioxidant Assay	IC_50_ (µg/mL)
DPPH	89.90 ± 12.80
ABTS	130.06 ± 18.25
Vitamin C (positive control)	7.02 ± 1.03

Data represent mean ± SD of triplicate determinations from three independent experiments.

## Data Availability

The datasets generated or analyzed during the present study are available from the corresponding authors.

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
