# Peer review of "Cytotoxic, Anti-Hemolytic, and Antioxidant Activities of Ruta chalepensis L. (Rutaceae) Extract, Fractions, and Isolated Compounds"

_plants, 2023, doi:10.3390/plants12112203_

Round 1
Reviewer 1 Report
Review comments to the author
Title: ''Antitumor, Anti-hemolytic, and Antioxidant Activities of Ruta chalepensis L. (Rutaceae) Extract, Fractions, and Isolated Compounds''.
Manuscript ID: plants-2399386.
1. Introduction:
- Page 2, Line 62: assign the type of hexane (n- or cyclo).
- Figure 1: The quality of the chemical structures should be improved.
2. Results and Discussion:
2.1. Phytochemical Data and Biodirected Isolation of Major Ruta chalepensis Compounds
- Page 3, Line 99: Add one space before citation [18].
3. Materials and Methods:
3.3. Plant Material and Extraction
- Page 6: The collected part should be mentioned.
- Page 6: The collection place should be mentioned.
- Page 6: The collection date should be mentioned.
- Page 6: The evaporation temperature should be mentioned.
- Page 6: Add the brand and model of the rotary evaporator.
Abbreviations:
- A list of abbreviations should be inserted by the end of the manuscript before references.
References:
- All the words ''In vitro'' should be written in italic font.
Supplementary Material:
- Page 2, Line 73: The number ''1'' after the carbon atom ''C'' in the formula ''C17H13NO3'' should be written in subscript font.
- The spectroscopic data (NMR) of the isolated compounds should be mentioned.
Minor editing of English language requiredز
Author Response
Dear reviewer, we appreciate the review of our work and all your comments; we look forward to answering all your questions. The document has been thoroughly revised following the suggestions of the individual reviewers.
The changes are marked with the change control tool, and the English language was also reviewed.
- Introduction:
- Page 2, Line 62: assign the type of hexane (n- or cyclo).
Reply.- Added. In this research, n-hexane was used.
- Figure 1: The quality of the chemical structures should be improved.
Reply.- Figures were improved
- Results and Discussion:
2.1. Phytochemical Data and Biodirected Isolation of Major Ruta chalepensis Compounds
- Page 3, Line 99: Add one space before citation [18].
Reply.- Done
- Materials and Methods:
3.3. Plant Material and Extraction
- Page 6: The collected part should be mentioned.
- Page 6: The collection place should be mentioned.
- Page 6: The collection date should be mentioned.
- Page 6: The evaporation temperature should be mentioned.
- Page 6: Add the brand and model of the rotary evaporator.
Reply.- These data have been added to the text.
Abbreviations:
- A list of abbreviations should be inserted by the end of the manuscript before references.
Reply.- The list of abbreviations has been added.
References:
- All the words ''In vitro'' should be written in italic font.
Reply.- All references were revised, and the corresponding italics were added.
Supplementary Material:
- Page 2, Line 73: The number ''1'' after the carbon atom ''C'' in the formula ''C17H13NO3'' should be written in subscript font.
Reply.- All formulas have been reviewed and corrected.
- The spectroscopic data (NMR) of the isolated compounds should be mentioned.
Reply.- The spectroscopic data have been added.
Reviewer 2 Report
Dear authors,
In my opinion the paper will improve if you apply the following:
1. Both anti-oxidant assays you used are non cell based (correct me if I am wrong). This should be corrected by applying for example the DCFH-DA assay
2. The MTT is a simple proliferation assay. You mention in your text that the extracts have anti-tumor properties. This is a very strong statement to make just by applying MTT. Anti-tumor effect may only be observed in vivo. Also other assays should be applied to support anti-cancer activity like cell cycle analysis, Annexin V/PI staining for apoptosis, changes in RNA/Protein levels to check for anticancer pathways etc....So if none of these are investigated, the extracts have anti-proliferative properties against cancer cells at best.
3. The cell lines you chose for MTT, are a cancer hepatocellular carcinoma cell line, a lymphoma cell line, PBMCs and a non-human normal cell line. This is not an optimal way to compare selectivity. I would suggest using more than one cancer cell line for each type of cancer. Also, a cell based anti-oxidant assay using the PBMCs would be preferable.
Author Response
Dear authors,
In my opinion the paper will improve if you apply the following:
Reply.- Dear reviewer, we appreciate the review of our work and all your comments; we look forward to answering all your questions. The changes made are marked with the change control tool.
- Both anti-oxidant assays you used are non cell based (correct me if I am wrong). This should be corrected by applying for example the DCFH-DA assay.
Reply.- We appreciate the recommendation. As you commented, the antioxidant activity assays were not cell-based; they were colorimetric assays. In addition, the rapid DPPH (PMID: 23790848) and ABTS (PMID: 27927090, PMID: 35158569) assays have proven to be appropriate, easy, and highly repeatable for the comparison of the antioxidant power of plant extracts as well as pure compounds (PMID: 23993591, PMID: 36760656, PMID: 35890424). Therefore, the antioxidant activity of the most effective treatment was determined by these methods, as mentioned in the paper.
- The MTT is a simple proliferation assay. You mention in your text that the extracts have anti-tumor properties. This is a very strong statement to make just by applying MTT. Anti-tumor effect may only be observed in vivo. Also other assays should be applied to support anti-cancer activity like cell cycle analysis, Annexin V/PI staining for apoptosis, changes in RNA/Protein levels to check for anticancer pathways etc....So if none of these are investigated, the extracts have anti-proliferative properties against cancer cells at best.
Reply.- Dear reviewer, the term antitumor has been changed to cytotoxic based on the literature (PMID: 34834687, PMID: 19501276), and in order to avoid the discrepancy between antitumor and antiproliferative activity (PMID: 37109486, PMID: 36771057). In addition to the above, in the next stage of research, we intend to determine the antitumor effect in In-Vivo assays, as well as the determination of the molecular mechanisms of action.
- The cell lines you chose for MTT, are a cancer hepatocellular carcinoma cell line, a lymphoma cell line, PBMCs and a non-human normal cell line. This is not an optimal way to compare selectivity.
Reply.- For the selectivity determination, it was decided to evaluate these cell lines due to their nature and based on the literature; VERO / HEP-G2 (adherent cells) (PMID: 29326018, PMID: 36365315), and PBMCs / lymphoma (non-adherent cells) (PMID: 37109486, PMID: 35055716). Information on this point has been added to the respective methodology.
I would suggest using more than one cancer cell line for each type of cancer. Also, a cell-based anti-oxidant assay using the PBMCs would be preferable.
Reply.- This recommendation, as well as the evaluation of the DCFH-DA assay, will be taken into account for the next stage of the research project.
Reviewer 3 Report
The study by Joel H. Elizondo-Luévano et al. entitled “Antitumor, Anti-hemolytic, and Antioxidant Activities of Ruta chalepensis L. (Rutaceae) Extract, Fractions, and Isolated Compounds” has been reviewed. The study aimed to evaluate some biological activities including the antitumor activity of R. chalepensis methanol extracts, sub-partitions, and major compounds.
The manuscript is potentially interesting, the rationale is clear, the methods and figures are exhaustive, and the main text needs substantial improvement.
Major
-Please rewrite the abstract clearly highlighting the results obtained. What about fractions and isolated compounds?
-In the introduction, please add more information on previous studies on Ruta chalepensis
-The results and discussions section is laconic in many parts and is a simple report of the obtained results. Results and discussions must be separated into two different sections and, the discussion must be conducted critically based on the results obtained and the scientific literature on the field such as PMID: 21434485; PMID: 28808378; PMID: 27729078; PMID: 24343999.
-The conclusions are misleading and confusing. Please clearly define the highlighted properties for the extract, fractions, and compounds and clarify the differences and the most promising properties. Clearly define if the most active is the phytocomplex within Ruta chalepensis or fractions/compounds and add your deductions.
Minor
-Keywords - Replace the words already present in the title
Line 49 - remove Fam: from “Ruta chalepensis L. (Fam: Rutaceae)”
Figure 1 - Please standardize the chemical structures of the reported compounds
Table 2 – In all the tables in the text add the correspondent letter to the significant differences p < 0.01 < 0.001 < 0.05 < 0.05 < 0.001 < 0.01
Author Response
The study by Joel H. Elizondo-Luévano et al. entitled “Antitumor, Anti-hemolytic, and Antioxidant Activities of Ruta chalepensis L. (Rutaceae) Extract, Fractions, and Isolated Compounds” has been reviewed. The study aimed to evaluate some biological activities including the antitumor activity of R. chalepensis methanol extracts, sub-partitions, and major compounds.
The manuscript is potentially interesting, the rationale is clear, the methods and figures are exhaustive, and the main text needs substantial improvement.
Reply.- Dear reviewer, we appreciate the review of our work and all your comments; we look forward to answering all your questions. The changes made are marked with the change control tool. As recommended by another reviewer, a list of abbreviations was added at the end of the paper, and the English language was also reviewed.
Major
-Please rewrite the abstract clearly highlighting the results obtained. What about fractions and isolated compounds?
Reply.- The abstract was improved, and the activities of the isolated compounds were highlighted.
-In the introduction, please add more information on previous studies on Ruta chalepensis
Reply.- The introduction was expanded, and data focused on the biological activity of R. chalepensis and the isolated compounds were added.
-The results and discussions section is laconic in many parts and is a simple report of the obtained results. Results and discussions must be separated into two different sections and, the discussion must be conducted critically based on the results obtained and the scientific literature on the field such as PMID: 21434485; PMID: 28808378; PMID: 27729078; PMID: 24343999.
Reply.- The comment was taken into account; the results and discussion sections were improved.
-The conclusions are misleading and confusing. Please clearly define the highlighted properties for the extract, fractions, and compounds and clarify the differences and the most promising properties. Clearly define if the most active is the phytocomplex within Ruta chalepensis or fractions/compounds, and add your deductions.
Reply.- The conclusions have been rewritten, focusing on the points indicated by the reviewer.
Minor
-Keywords - Replace the words already present in the title
Reply.-Done
Line 49 - remove Fam: from “Ruta chalepensis L. (Fam: Rutaceae)”
Reply.- Done
Figure 1 - Please standardize the chemical structures of the reported compounds.
Reply.- Figures were improved
Table 2 – In all the tables in the text, add the correspondent letter to the significant differences p < 0.01 < 0.001 < 0.05 < 0.05 < 0.001 < 0.01
Reply.- Significances have been added.
Round 2
Reviewer 2 Report
No further comments
Reviewer 3 Report
The resubmitted manuscript has been significantly improved and is potentially interesting, the rationale is now clear, and the flow is better structured. So, overall I am positive about the possible publication in the journal.